systems biology

precancer, ageing, squamous epithelia, tissue homeostasis, statistical approaches

**Authors for correspondence:**
Philip H. Jones
e-mail: pj3@sanger.ac.uk
Benjamin A. Hall
e-mail: b.hall@ucl.ac.uk

# Methods for analysing lineage tracing datasets

Vasiliki Kostiou[1], Huairen Zhang[2], Michael W. J. Hall[2,3], Philip H. Jones[2,3] and Benjamin A. Hall[1]

[1]Department of Medical Physics and Biomedical Engineering, UCL, Gower Street, London WC1E 6BT, UK
[2]MRC Cancer Unit, University of Cambridge, Hutchison-MRC Research Centre, Box 197, Cambridge Biomedical Campus, Cambridge CB2 0XZ, UK
[3]Wellcome Trust Sanger Institute, Hinxton CB10 1SA, UK

MWJH, 0000-0003-2904-6902; BAH, 0000-0003-0355-2946

A single population of progenitor cells maintains many epithelial tissues. Transgenic mouse cell tracking has frequently been used to study the growth dynamics of competing clones in these tissues. A mathematical model (the 'single-progenitor model') has been argued to reproduce the observed progenitor dynamics accurately. This requires three parameters to describe the growth dynamics observed in transgenic mouse cell tracking—a division rate, a stratification rate and the probability of dividing symmetrically. Deriving these parameters is a time intensive and complex process. We compare the alternative strategies for analysing this source of experimental data, identifying an approximate Bayesian computation-based approach as the best in terms of efficiency and appropriate error estimation. We support our findings by explicitly modelling biological variation and consider the impact of different sampling regimes. All tested solutions are made available to allow new datasets to be analysed following our workflows. Based on our findings, we make recommendations for future experimental design.

## 1. Introduction

The progenitor cell dynamics of squamous epithelial cells is a major subject of study in biomedicine. Squamous epithelial tissues cover the external surface of the body, the mouth and the oesophagus. Importantly, most common human cancers develop from these tissues. Understanding the rules of cell fate decision is therefore fundamental to explain not only healthy tissue growth and maintenance but also the mechanisms of wound healing, mutagenesis and cancer. Epithelial tissues consist of layers of keratinocytes, and in mice, skin and oesophageal tissues are maintained by a single layer of cells at the base of the tissue. Progenitor cells in this basal layer stochastically differentiate, cease cell division, and then stratify into the upper layers of the tissue, migrating to the surface before eventually

**Figure 1.** The architecture and maintenance of murine stratified squamous epithelial tissues. (*a*) Proliferation is restricted to the deepest basal layer. Upon differentiation, basal cells exit the cell cycle and migrate through suprabasal layers, until eventually, they reach the surface where they are shed from the tissue. Cell production and loss should be perfectly balanced so that homeostasis and healthy function is achieved. (*b*) According to the single-progenitor model, stratified epithelial tissues are maintained by a single, equipotent population of progenitor cells which divide stochastically to generate either two proliferating daughters, two differentiating daughters, or one daughter of each type.

being shed (figure 1*a*). Human squamous tissues have a more complex organization but share several features and are believed to be maintained in a similar manner.

The 'single-progenitor' (SP) model (figure 1*b*) has been shown over several studies to accurately describe the observed progenitor cell dynamics in transgenic lineage tracing experiments [1–7]. Using three parameters, the division rate, the stratification rate and the probability of symmetric division, this model predicts average clone size, clone size distributions, tissue homeostasis and clone survival probabilities. While simulation-based techniques have been used to fit experimental data, an analytical solution has also been described [8] allowing for maximum-likelihood calculations. Model parameterization and accurate representation of uncertainty, however, remains problematic, particularly in light of short-term data from *in vivo* histone dilution assays and live imaging that show cell cycle distribution times do not follow the exponential distributions assumed in the analytical model [6,9].

Here, we report the results of exploring alternative approaches applied to the analysis of both published and synthetic datasets. We find that simulation-based maximum-likelihood methods require extensive sampling to find a distribution of parameters making them intractable for many analyses. We further find that the use of the published analytical solution allows the identification of a single parameter with narrow confidence intervals. However, the analysis of synthetic datasets with realistic cell cycle distribution times and biological variation between samples suggests that these intervals are too narrow and do not accurately reflect uncertainties in the method and underlying data. We go on to show that an approximate Bayesian computation (ABC)-based approach using a non-Markovian simulator gives appropriate error bars at an acceptable computational cost. All code underlying this is made available as a Python notebook, enabling the easy analysis of newly collected datasets. Finally, we use synthetic data to explore the relationship between parameterization and the methods of data collection, concluding that single timepoints with typical sampling from the literature (three mice) show high variability that when analysed in isolation could be open to misinterpretation.

## 2. Results

### 2.1. Approaches based on approximate Bayesian computation more accurately estimate parameters and uncertainties than maximum-likelihood

In order to identify the most appropriate strategy for parameterizing progenitor cell clonal dynamics, we explored the effectiveness of different inference techniques. Maximum-likelihood approaches have been

widely used in previous publications [3,4,10]. In this approach to the analysis of transgenic lineage tracing, the likelihood of different parameter combinations is estimated from the frequency at which different clone sizes have been observed at different timepoints and a calculation of the probability of a given clone size arising. This probability can be calculated using a published analytical solution of the branching process describing the SP model [8]. This, however, assumes that cell cycle times are exponentially distributed, which is not the case and can undermine this analysis [9]. In this situation, bespoke simulation tools need to be run with extremely high sampling (over 100 000 simulations per parameter combination) in order to calculate the clone size probability distributions accurately, at a substantial computational cost. The estimates of likelihood calculated from the observed clone sizes and probability distributions can then be used to calculate the most likely combination of parameters and confidence limits.

An alternative approach, ABC, does not rely upon the calculation of the probability of clone sizes. Instead, models with specific parameter sets are simulated and the outputs compared with experimental observations. As this approach is simulation-based, it is insensitive to the use of non-Markovian processes. Using a distance metric such as the inter-quantile distances between distributions, or the Kolmogorov–Smirnov (KS) statistic, sets of parameters can be collected that have similar properties to the experimental observations. Sequential Monte Carlo approximate Bayesian computation is an ABC protocol that can efficiently identify parameters consistent with experimental observations. Briefly, this approach takes an initial set of parameters, selected from a user chosen prior, and perturbs and then simulates observational data from them. Individual parameters are accepted or rejected based on a threshold applied to a summary statistic that compares simulated and observed data. This process is iterated with a progressively reduced threshold. Through repeated rounds of testing, perturbation and rejection populations of parameters can be identified that fit the data increasingly well, and can be used to identify both uncertainty and best-fitting parameters [11,12]. This approach is continued until the rejection rate starts to rise, at which point, it is believed that overfitting starts to occur [11]. One advantage of SMC-ABC over maximum-likelihood estimation (MLE) approaches is in the flexibility and efficiency of the approach, as it obviates the need for analytical solutions or large-scale simulations.

The above three methods (MLE with analytical solution, MLE with simulation and ABC) were tested on the analysis of both experimental [3] and synthetic datasets with exponentially distributed cell cycle times. These simulated datasets were generated based on the clonal data provided in [3] and allowed us to use an increased sampling (100 000 total simulated clones) compared to the typical sampling followed in the experimental protocols (three animals per timepoint, 100 clones per animal). All approaches broadly agreed on the estimated parameters for both experimental (figure 2$a$–$c$) and synthetic data (figure 2$d$–$f$). MLE using the analytical solution was able to infer the expected parameter values efficiently ($r = 0.064$, $\rho = 0.68$), producing narrow confidence intervals ($r = (0.05, 0.068)$ 95% CI, $\rho = (0.64, 0.72)$ 95% CI) and a smooth likelihood distribution when applied to the experimental dataset (figure 2$a$). By contrast, MLE using model simulations suggested that a sample size larger than 100 000 clone simulations per parameter set would be required to produce an appropriate distribution, increasing computational effort substantially. Simulations of 100 000 clones for $19 \times 19$ parameter combinations estimated similar parameter values ($r = 0.06$, $\rho = 0.64$) but produced unrealistically narrow confidence intervals of zero width. As the shape of the likelihood distribution was no longer smooth, this suggested that sampling undermined the analysis (figure 2$b$). The increased computational demand required for this analysis (28 h CPU time for $19 \times 19$ combinations compared to 1.3 h using the analytical solution for the same grid) restricted the number of parameter combinations that were searched.

In contrast with both MLE approaches, the SMC-ABC approach produced a smooth distribution (figure 2$c$) with substantially larger confidence intervals when used to analyse the experimental data ($r = 0.06$ (0.04, 0.073) 95% CI, $\rho = 0.71$ (0.56, 0.77) 95% CI). While the peaks of the MLE analysis were within these intervals, the distribution of acceptable parameters was offset relative to the MLE likelihoods. Given that it is known that there is substantial biological variation (division times are estimated to vary by around 10% [9]), this raises the question of whether MLE approaches were underestimating the uncertainty that arises due to biological variation. Alternatively, the analysis of synthetic data, which is based on a single parameter, and hence has no uncertainty from biological variation (figure 2$d$–$f$), may suggest that uncertainty is overestimated in ABC, where broad CI were observed. Furthermore, a known issue of the analytical solution is the assumption of an exponential distribution of progenitor cell cycle times. Both histone dilution and live imaging experiments on epithelial tissues have shown that there is a refractory period in which no division can occur [6,9].

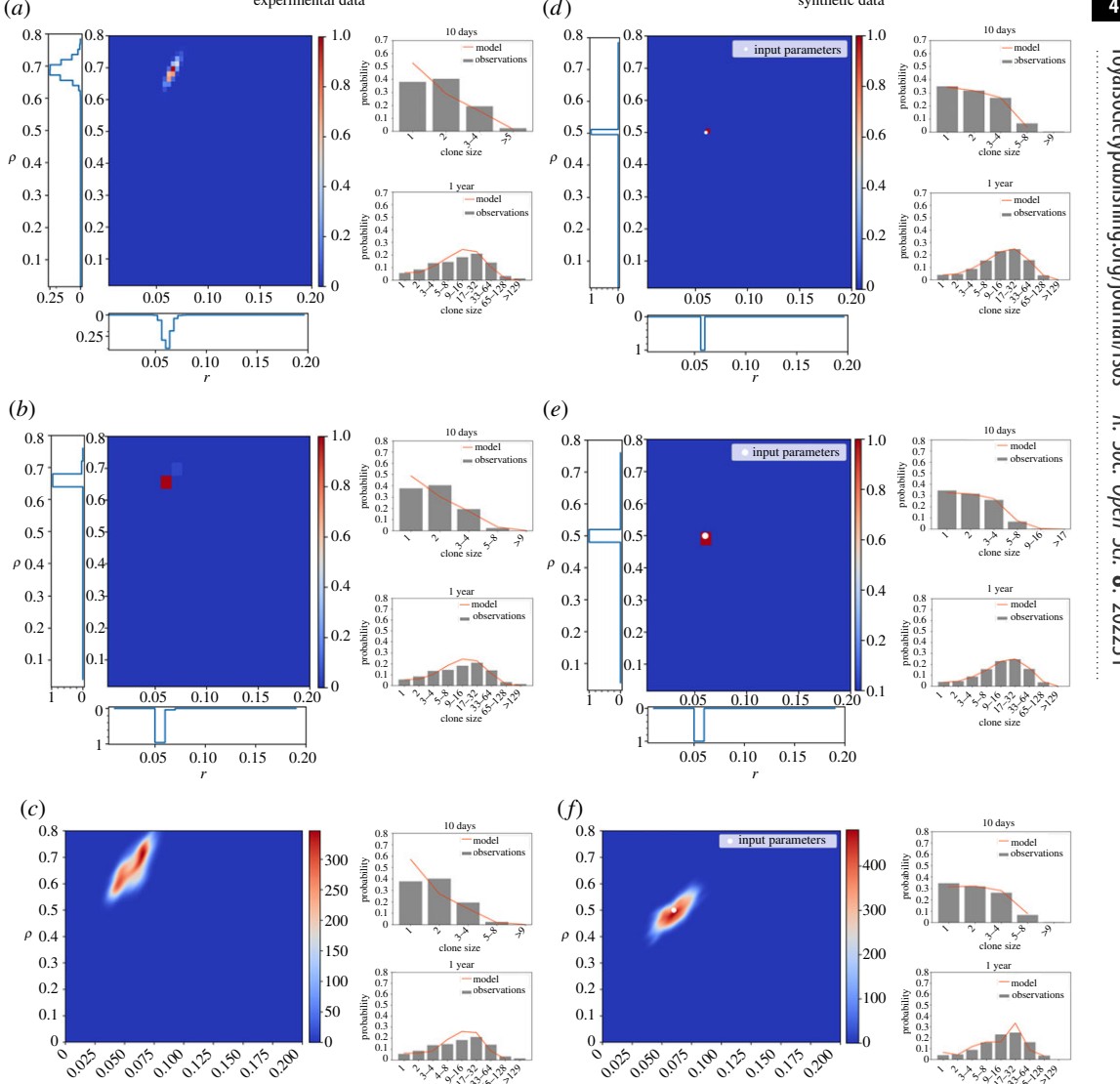

**Figure 2.** Analytical solution and simulation-based methods inferring single-progenitor parameters from both lineage tracing and synthetic datasets. The three different parameter estimation strategies applied to mouse oesophagus lineage tracing data from [3] (a–c) and synthetic datasets generated by performing SP model simulations with parameters $\lambda = 2.9$/week, $r = 0.06$, $\rho = 0.5$ and assuming exponentially distributed cell cycle time (d–f). (a,d) A maximum-likelihood approach based on the analytical approach gives a narrow distribution of likelihoods for each parameter. (b,e) With 100 000 simulations per parameter set, inferred likelihoods are noisy and fail to give smooth distributions in $r$ or $\rho$. (c,f) SMC-ABC running for 10 generations. Heatmap plots Kernel Density Estimate of the final population of parameter sets. a,b,d,e: Heatmap shows the likelihood distribution, while likelihood estimates for each parameter are shown alongside. (a–f) Right: Basal clone size probability distributions of the input data (grey) and the inferred parameters (orange) as obtained at an early and late timepoint.

This can undermine the analysis leading to the identification of incorrect parameters [9]. The issue in assuming an underlying exponentially distributed cell cycle time is also highlighted in the obtained clone size distributions at early time points, where the difference to the clone size distributions of the experimental data is higher compared to later time points (figure 2a–c; electronic supplementary material, figures S1 and S2).

To test the influence of both cell cycle distribution times, and biological variation, both features were included in models used to generate a new set of synthetic data. These new synthetic datasets were generated with Gamma distributed cell cycle times (Gamma distribution parameters are taken from [9]). Furthermore, $r$, $\rho$ and $\lambda$ parameter values were drawn from a normal distribution around $\lambda = 2.9$/ week ± 0.1 s.d., $r = 0.09 ± 0.01$ s.d., and ρ = 0.7 ± 0.05 in order to introduce noise and mimic the variation observed in real datasets. For this analysis, we chose parameter values in broad agreement

with the previously inferred values. Where the analysis allowed, we took account of the shape of the gamma distribution. This can be inferred through an orthogonal experimental protocol and so was not fit alongside other parameters [9].

Application of the analytical solution to the realistic synthetic datasets revealed that the estimated likelihoods were both inaccurate and overly precise (figure 3a). The introduction of more non-Markovian cell cycle times leads to misleading calculated parameters in the analytical MLE approach (figure 3a). Moreover, the narrowness of the confidence intervals calculated with this method excludes the true parameters. We suggest that this underestimation of uncertainty arises as a result of limited sampling, and this same issue has been noted in other, unrelated systems [13,14].

In contrast with the analytical approach, simulation-based techniques have the flexibility to account for more realistic cell cycle times. Simulation-based MLE accurately calculated the input parameters and successfully reproduced the expected clone size distributions (figure 3b, electronic supplementary material, figure S3). However, the confidence intervals do not reflect the biological variation in the samples, substantially underestimating the uncertainty in $r$ and $\rho$ (figure 3b).

To address this, a SMC-ABC approach using a non-Markovian simulator was applied. We find that this generates a broad, smooth distribution that reflects the input parameters and uncertainty (figure 3c; electronic supplementary material, figure S3). Compared with results from MLE approaches, these findings demonstrate that the SMC-ABC inference technique is the most appropriate method for analysing lineage tracing datasets, while also being substantially more efficient.

## 2.2. The contribution of individual timepoints to the likelihood distribution is sensitive to both the specific time and biological variation

Previous studies have used the results from individual timepoints to make arguments about the selection and rejection of different models [5]. The underestimation of noise by MLE approaches raises the question of how biological variation influences the distributions observed at each individual timepoint. Related to this is the question of how each timepoint contributes to the wider likelihood. Answering these questions would aid experimental interpretation, confirming whether individual timepoints should be studied in isolation, but also offer a route to optimizing experimental design.

To explore these questions, we revisited the experimental datasets and used synthetic datasets with modelled biological variation and calculated the likelihood distributions. As we are primarily interested in the shape of the distributions rather than the specific parameters proposed, we used the analytical MLE approach, applied to synthetic data with an exponential division time distribution. In order to explore how likelihood distributions vary with individual timepoints, we chose this approach as we expect the shape to be estimated accurately. The parameter likelihood distributions calculated from individual timepoints vary strongly with time and have extremely broad distributions, frequently covering large parts of parameter space. As such they are insufficient to estimate parameters in isolation (figure 4a). We further conclude that the precise estimation of likelihood from the whole timeseries effectively arises from the overlap of these distinct, broad distributions—that is to say, individual timepoints do not strongly point to a small distribution of parameters, but when considered together, they only agree on a narrow set.

When compared with likelihood distributions calculated from experimental measurements, while we find that while the distributions are generally highly similar, the small number of biological replicates at each timepoint (two or three) leaves them prone to distortion by chance parameter combinations (figure 4a, 180 days). This would suggest that individual timepoints should not be used to draw strong conclusions as they are highly prone to chance variation.

One feature of note is that while the likelihood varies over time, later timepoints become increasingly similar. This raises the question of whether all timepoints need to be collected, and whether some timepoints could be omitted to increase sampling at other timepoints, potentially reducing the cost of the experiment and making individual timepoints more reliable. To investigate this, parameter likelihoods were calculated for synthetic datasets with increased sampling per timepoint (figure 4b). We then performed subsequent likelihood estimations considering different timepoint combinations (figure 4c–e), while maintaining the total number of synthetic 'mice' considered. We find that including just early (figure 4c) or late (figure 4d) timepoints were insufficient to estimate the correct parameters of the synthetic datasets. Late timepoints illustrate some of the key properties of the system, such as the linear growth of average clone sizes over time, which are key pieces of evidence in favour of the SP model. Early timepoints in contrast have distinct likelihood distributions,

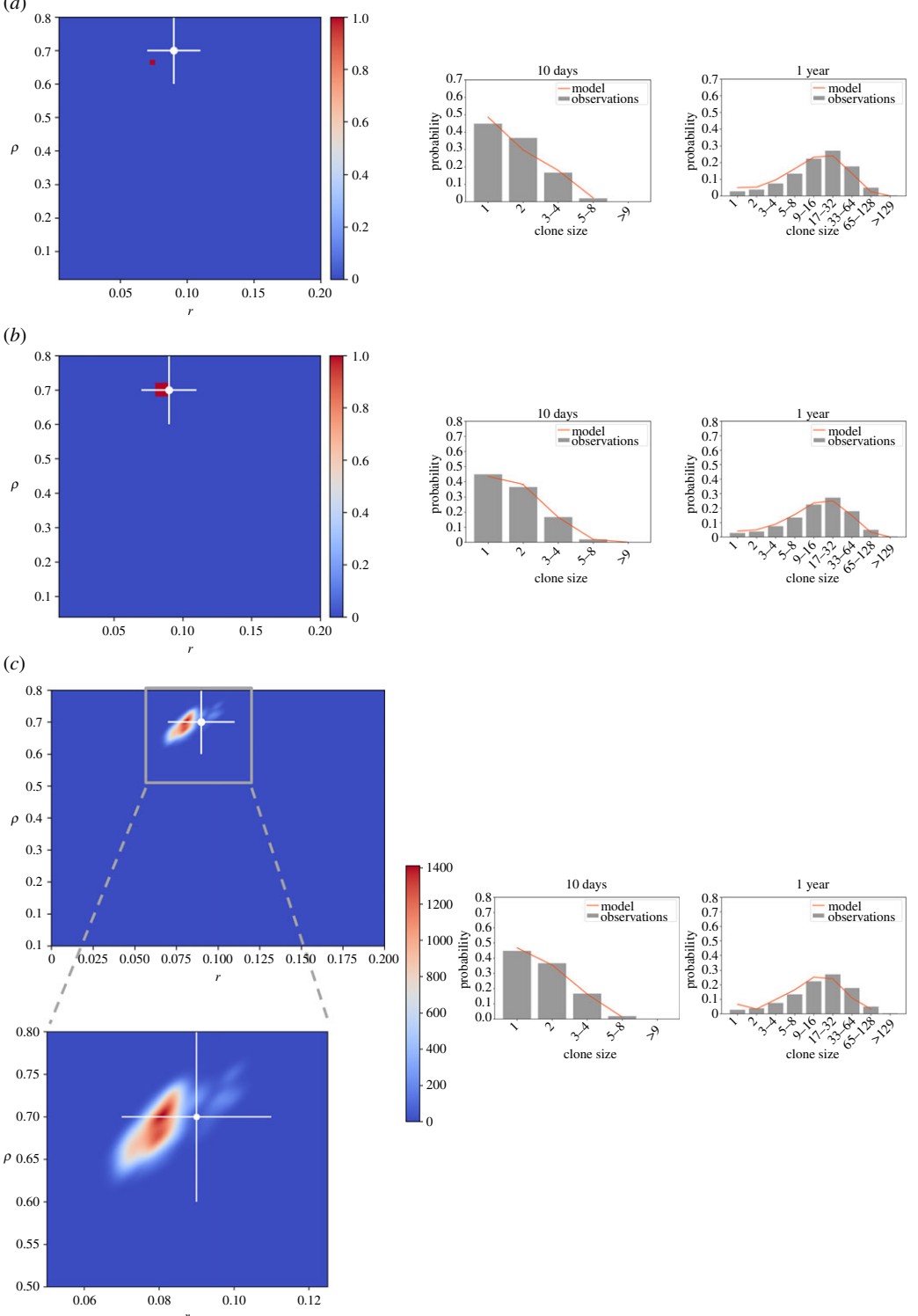

**Figure 3.** ABC approaches efficiently and appropriately parameterize the single-progenitor model from lineage tracing data. (*a*) Synthetic datasets where realistic cell cycle distribution times and biological variation are modelled show that likelihoods calculated from the analytical engine are overly conservative and inaccurate. Heatmap plots likelihood distribution calculated using the analytical solution. (*b*) MLE-based simulations considering non-Markovian cell cycle distribution times improve parameter estimation but still fail to give smooth likelihood distributions. Heatmap plots likelihood distribution. (*c*) An ABC approach to inferring parameters gives a smooth distribution and reasonable confidence intervals. Heatmap plots Kernel Density Estimate of the final population of parameter sets. A zoomed version of the plot is shown at the bottom. (*a*–*c*) Input parameters and error estimates indicated by cross and error bars (2 x s.d.). Synthetic datasets were generated using a mean *r*: 0.09 ± 0.01 s.d., $\rho$: 0.7 ± 0.05 s.d., $\lambda$: 2.9 ± 0.1 s.d. and assuming Gamma distributed cell cycle times. Right: Basal clone size probability distributions of the input synthetic data (grey) and the inferred parameters (orange) as obtained at an early and late timepoint.

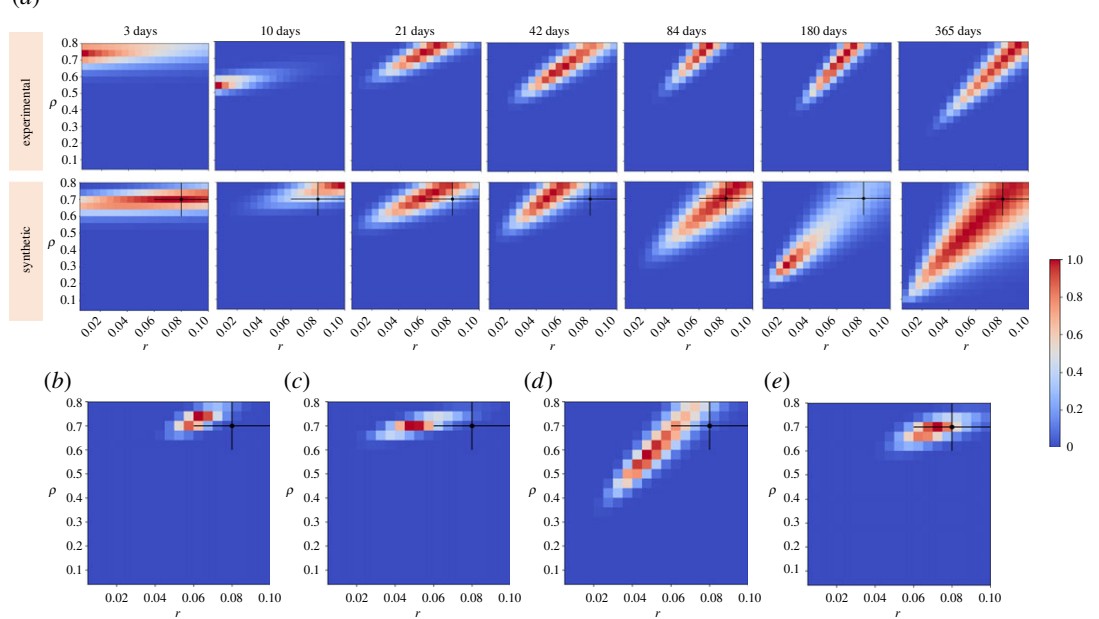

**Figure 4.** Estimated likelihood distributions across individual timepoints are highly sensitive to biological variation. (*a*) Parameter likelihood distributions across individual timepoints calculated by the analytical engine on mouse oesophagus lineage tracing data from [3] (top) and synthetic datasets with biological variation (bottom). Likelihoods from individual timepoints are insufficient to estimate parameters on their own. (*b*–*e*) Inferred parameter likelihoods estimated by the analytical engine on synthetic data with biological variation and increased sampling (five mice per timepoint) considering all timepoints: 3, 10, 21, 42, 84, 180 and 365 days (*b*), the three earliest timepoints: 3, 10 and 21 days (*c*), the three latest timepoints: 84, 180 and 365 days (*d*), a combination of early, middle and late timepoints: 3, 42 and 365 days (*e*). (*a*–*e*) Synthetic datasets were generated using a mean $r$: $0.08 \pm 0.02$ s.d., $\rho$: $0.7 \pm 0.1$ s.d., $\lambda$: $2.85 \pm 0.15$ s.d. and assuming exponentially distributed cell cycle times. Input parameters with their standard deviations are indicated by a cross.

supporting parameter inference. Considering the above points, we investigated a combination of one early, middle and late timepoint which we found to be sufficient to estimate the expected likelihood distributions successfully (figure 4*e*).

# 3. Discussion

Epithelial tissue maintenance is particularly important to understand homeostasis and processes such as ageing, preneoplasia and cancer formation. The advent of genetic lineage tracing has provided useful insights into epithelial progenitor cell fate decision processes. Such studies are increasingly popular and this underlines the need for appropriate tools to analyse such datasets. Progenitor cell dynamics in multiple epithelial tissues has been argued over several years to be described by the SP model [1–4,6], a mathematical model described with only three parameters, the division rate ($\lambda$), a stratification rate ($\Gamma$) and a probability of symmetric division ($r$). Despite this simplicity, we showed here how the choice of statistical approach can undermine the analysis, and how simulated datasets can be used to validate the approach taken. A key finding was that maximum-likelihood calculations based on either an analytical approach [8] or simulation was overly precise, and produced unrealistic estimations of uncertainty that had the potential to obscure true parameters. In extremis, when tested on synthetic datasets that explicitly accounted for realistic cell cycle distributions and inter-mice biological variation, the proposed parameter values derived from the analytical solution were both inaccurate and the true values were outside confidence intervals. Simulation-based maximum-likelihood approaches were less inaccurate than the analytical solution, but had similar issues with uncertainty, and their utility was limited by their computational cost. In addition to the supercomputing power required for this analysis, the requirement to perform large numbers of simulations additionally limited the granularity of the analysis, effectively reducing the number of parameters that were tested. We found that despite being substantially less intensive, the SMC-ABC approach was able to account for realistic cell cycle times and identify the input parameters accurately with realistic error estimates.

The accurate estimation of uncertainties in model parameters is important in a wide range of fields. In the field of progenitor cell homeostasis specifically, there are active debates around the fundamental processes that underpin maintenance of different tissues, and overconfidence in apparent parameters could mislead researchers. Additionally, while there are multiple techniques that exist to explore progenitor cell dynamics in the tissue, their analysis frequently leads to different parameter estimates. In this context, an accurate treatment of uncertainty could aid the comparison between the outcomes of different experimental methods and historical results. This is important in the light of discrepancies in parameter estimation between lineage tracing and live imaging approaches [4]. Furthermore, while the accurate estimation of uncertainties leads us to wider confidence intervals, it still allows us to reject a random cell fate as proposed by [15].

The tool presented here and distributed with this manuscript, allows for accurate and efficient analysis of newly collected datasets following our protocol. It should be noted that while this tool enables the analysis of epithelial tissues, it can also be applied in other systems where cohesive clones are observed. Our method could also be applied to tissues that are maintained by alternative stem cell/progenitor cell dynamics, after modifications to the code to reflect the alternative model.

A major motivation for refining experimental design is to maximize the information generated while reducing costs and in particular reducing the need for animal use. Our findings suggest that increasing sampling at individual timepoints while reducing the total number of timepoints would increase the reliability of individual timepoints without impacting parameter estimation. While this has implications on experimental design in transgenic systems, it also illustrates how simulation and modelling can aid experimental design beyond statistical tests. Here, simulating clone growth enables us to efficiently explore how parameters may be measured and the implications of well-known confounders like inter-mouse variation on what can be safely interpreted from a dataset. The adoption of similar approaches can be both used before the experiment is run, to establish a protocol, but also afterwards to confirm whether unexpected features of the data can be trusted.

# 4. Methods

## 4.1. The single-progenitor model

The SP model states that the tissue is maintained by a single, equipotent progenitor population of basal cells that are able to give rise to either progenitor cells or differentiating daughters stochastically. The progenitor cell compartment in the basal layer is modelled as containing a mixture of progenitor cells, which go on to divide, and differentiated cells, which go on to stratify into upper layers of the tissue. Cell fate is determined on cell division; a progenitor cell divides to either produce one differentiated and one progenitor cell (asymmetric division), or either a pair of differentiated cells or a pair of progenitor cells (symmetric division) (figure 1b). This model of tissue renewal can be described as a continuous-time Markovian process, as shown by [1,3] equation (4.1):

$$
\begin{aligned}
A \xrightarrow{\lambda} &\begin{cases} AA & r \\ AB & 1-2r \\ BB & r \end{cases} \\
B &\xrightarrow{\Gamma} C \\
C &\xrightarrow{\mu} \emptyset,
\end{aligned}
\tag{4.1}
$$

where $A$ represents the basal layer progenitor cells, $B$ the basal cells committed to differentiate and $C$ the suprabasal layer cells. Progenitor cells divide regularly with an overall division rate $\lambda$ and give rise to either two progenitor daughters ($AA$), two differentiating daughters ($BB$) or one daughter of each type ($AB$) with fixed probabilities. Given the fact that $AA$ symmetric division leads to clone expansion and $BB$ symmetric division tends towards clone extinction, the two symmetric division rates should be equal in order for a steady state in terms of the number of cells to be maintained across the progenitor clone population. The probabilities of symmetric and asymmetric divisions are $r$ and $1-2r$, respectively, with $0 < r \leq 0.5$. Differentiating daughters in the basal layer stratify to the suprabasal layer at rate $\Gamma$ and suprabasal cells, $C$, are shed at rate $\mu$. The fraction of cells in the basal layer that goes on to divide is $\rho$. As homeostasis dictates that the total basal layer cell population stay constant over

time, the stratification rate can be calculated from other parameters, as shown by [1,3]:

$$\Gamma = \frac{\rho}{1-\rho}\lambda.$$

Therefore, parameterizing the model requires specific values to be found for $r$, $\lambda$ and $\rho$.

## 4.2. Generating clone size distributions following the single-progenitor model

As the SP model obeys a continuous-time Markov process, the time evolution of proliferating ($A$) and differentiating ($B$) cell populations can be formulated in terms of the stochastic Master equation:

$$\frac{dP_{n_A,n_B}}{dt} = r(n_A - 1)P_{n_{A-1},n_B} + (1 - 2r)n_A P_{n_A,n_{B-1}} + r(n_A + 1)P_{n_{A+1},n_{B-2}}$$
$$+ \Gamma(n_B + 1)P_{n_A,n_{B+1}} - (n_A + \Gamma n_B)P_{n_A,n_B}, \tag{4.2}$$

where $P_{n_A, n_B}$ denotes the probability of finding clones containing $n_A$ proliferating cells and $n_B$ differentiated cells. Antal and Krapivsky sought to provide an exact formula for the Master equation, calculating clone size probabilities, $P_{n_A, n_B}(t)$, and clone survival probabilities. The exact solution was obtained as shown in equation (4.3) (the full derivation is given in [8]).

$$F = 1 - u + \frac{u(1 + v) - \gamma(1 + 2w)}{2r} + \frac{\gamma}{2r}\frac{(1 + 2w)M_{1+w,0}(g) - 2CW_{1+w,0}(g)}{M_{w,0}(g) + CW_{w,0}(g)}$$
$$g = \frac{uv}{\gamma}, \quad v = \sqrt{1 - 4r}, \quad w = \frac{\gamma(1 - 2r) - 2r}{2\gamma v}, \quad u = (1 - y)e^{-yt} \tag{4.3}$$

C is a constant determined as shown in equation (4.4):

$$C = \frac{-\theta M_{w,0}(\hat{g}) + (1 + 2w)M_{1+w,0}(\hat{g})}{\theta W_{w,0}(\hat{g}) + 2W_{1+w,0}(\hat{g})},$$
$$\text{where} \quad \theta = 1 + 2w - \hat{g} + \frac{2r(x - y) + y - 1}{\gamma}, \quad \hat{g} = \frac{(1 - y)v}{\gamma}. \tag{4.4}$$

The analytical formula relies upon the use of confluent hypergeometric functions (Whittaker functions denoted by the terms $W$ and $M$). To use the analytical solution in our analysis, we implemented the analytical formula in Python. Specifically, for every $r$, $\rho$ combination ($0 < r < 0.5$ and $0 < \rho < 1$) and for every time point equation (4.3) was called to compute the probability for a given basal clone size $n$. Clone size probabilities were searched for a set of $49 \times 49$ $r$ and $\rho$ parameter combinations. As $\lambda$ was measured independently from H2BGFP dilution assays and provided to us, we used a fixed $\lambda$ value when computing clone size probabilities for both experimental and synthetic datasets.

As an alternative to the analytical solution, the time evolution of clonal populations following the SP paradigm was also simulated using the Gillespie stochastic simulation algorithm [16]. Specifically, for a fixed $\lambda$ value and for $19 \times 19$ $r$, $\rho$ parameter combinations ($0 < r < 0.5$ and $0 < \rho < 1$), multiple Gillespie simulation repetitions were performed ($N = 100\,000$) to estimate the probability of observing a given basal clone size $n$ at a given time point $t$. Initially, SP model simulations were performed assuming exponentially distributed cell cycle times. Non-Markovian simulations of the SP model were performed assuming a Gamma-shaped cell cycle time distribution [9].

Synthetic datasets were initially generated by performing multiple Gillespie SP model simulations under a specific $\lambda$, $r$, $\rho$ parameter set. Synthetic datasets with biological variation and realistic cell cycle times were generated by performing simulations that took into account the number of mice typically used in a lineage tracing experiment (two to three per timepoint). Specifically, we performed multiple Non-Markovian SP model simulations ($N = 10\,000$), considering seven different timepoints. This process was repeated 21 times, therefore simulating 21 animals so that three mice were considered per each time point. The simulation set corresponding to an individual mouse was assigned slightly different $\lambda$, $r$, $\rho$ values drawn from a normal distribution to include inter-mice variability ($\lambda = 2.9 \pm 0.1$, $r = 0.09 \pm 0.01$, $\rho = 0.7 \pm 0.05$).

## 4.3. Single-progenitor parameter inference

To fit the SP model simulations against clonal datasets and identify appropriate parameter values, the following inference techniques were tested. Fitting was performed on parameters $r$ and $\rho$.

### 4.3.1. Maximum-likelihood estimation

To calculate the likelihoods for the SP model parameters, a grid search was performed on a range of valid parameter values ($0 < r < 0.5$ and $0 < \rho < 1$) and the theoretical estimates of basal clone size distributions—obtained either analytically or by performing stochastic Gillespie simulations—were contrasted with the ones observed experimentally by assessing the log-likelihood of every parameter combination $\theta$. The most probable parameter combination was then selected as the parameter set with the maximum log-likelihood,

$$l(\theta;\ x) \ = \ \sum_t \sum_n (x_n(t) * \log p_n(t,\ \theta)), \tag{4.5}$$

where $x_n(t)$ corresponds to the frequency of measured clone sizes with n basal cells at time $t$ and $p_n(t, \theta)$ is the probability of observing clones of size $n$ at time $t$ for a given parameter set values $\theta$.

### 4.3.2. Sequential Monte Carlo approximate Bayesian computation

To infer the parameters for the SP model an SMC-ABC approach was followed [11,17]. Simulations of the SP model were performed starting from initial $r$ and $\rho$ values ($0 < r < 0.5$ and $0 < \rho < 1$) drawn from a uniform distribution, used as prior. For every simulation round (population), a distance metric was computed for every value pair based on the sum of the KS's test distance summary statistic.

While iterating over successive populations ($N = 10$), the new parameter sets to be tested were derived from a resampled and perturbed weighted set of points previously drawn. Perturbation allows a more efficient exploration of the parameter space. Parameter values with calculated distance above a certain threshold (tolerance) were rejected, thus aiming to obtain the posterior distribution after several rounds. Tolerance is decreased after each round. The population size (number of particles to be accepted at each round) was set to 500.

The runtimes of the different analysis workflows were the following: the total CPU time for the analytical solution with MLE considering a $49 \times 49$ grid of parameter values was 10.2 h. The total CPU time for the MLE-based simulations considering 100 000 clone simulations per $19 \times 19$ combinations of parameters was 28 h. The total CPU time for the SMC-ABC considering 10 populations, a population size of 500 and simulating 1000 clones per parameter combination was 38.33 h. All analyses ran on a single core of an Intel(R) Xeon(R) processor (E5-2698 v3 @ 2.30 GHz).

All code was implemented in python 3.6, in Jupyter notebooks, using numpy. SMC-ABC was used through the pyABC library. Code is available in electronic supplementary material.

Data accessibility. Models are included in this submission as a compressed zip, attached as electronic supplementary material.

Authors' contributions. V.K. ran calculations, wrote and reviewed code, planned the simulations and wrote the manuscript. H.Z. performed simulations and wrote code. M.W.J.H. performed code review and wrote the manuscript. P.H.J. wrote the manuscript. B.A.H. wrote and edited the manuscript, supervised and designed the study.

Competing interests. We declare we have no competing interests.

Funding. This work has been supported by the Royal Society (URF to B.A.H. grant no. UF130039, studentship to V.K.). B.A.H. is supported by a Medical Research Council (MRC) Grant-in-Aid to the MRC Cancer unit. M.W.J.H. acknowledges support from the Harrison Watson Fund at Clare College, Cambridge. P.H.J. is supported by a Cancer Research UK Programme grant no. (C609/A17257) and core grants from the Wellcome Trust to the Wellcome Sanger Institute, 098051 and 206194.

Acknowledgements. We thank Allon Klein, and the Hall and Jones groups for useful discussions.

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
