## [Peer Review File · Royal Society Open Science]

Review History

RSOS-202231.R0 (Original submission)

Review form: Reviewer 1

Is the manuscript scientifically sound in its present form?

Yes

Are the interpretations and conclusions justified by the results?

Yes

Is the language acceptable?

Yes

Do you have any ethical concerns with this paper?

No

Have you any concerns about statistical analyses in this paper?

No

Recommendation?

Accept with minor revision (please list in comments)

Comments to the Author(s)

This work compares three different methods of determining the parameters of the "single progenitor model" from clone size distributions. The authors use both published and synthetic data to argue that a Bayesian-based method is optimal in terms of time and accuracy.

The problem is very interesting and timely, and the results are convincing and relevant. The manuscript is very well written, all algorithms, assumptions, and conclusions are clearly stated. I therefore recommend the manuscript for publication following a minor review.

Questions:

- Maximum likelihood method is in principle Bayesian inference with uniform prior. Could the authors briefly comment on the fundamental difference between the MLA and SMC-ABC approaches described in the manuscript?
- Jupyter notebook: I managed to get the code running, but the run time is very long and consequently it is not easy to explore the meaning of various parameters in the code. Would it be possible to add a simple demonstration that would run no longer than a minute?

Review form: Reviewer 2

Is the manuscript scientifically sound in its present form?

Yes

Are the interpretations and conclusions justified by the results?

Yes

Is the language acceptable?

Yes

Do you have any ethical concerns with this paper?

No

Have you any concerns about statistical analyses in this paper?

No

Recommendation?

Accept with minor revision (please list in comments)

Comments to the Author(s)

In this paper the authors explore alternative strategies for the analysis of clonal lineage tracing data focussing on the previously described committed progenitor cell model. They use a combination of experimental and simulated data to test the accuracy and efficiency of three approaches allowing them to make recommendations for future analyses. My expertise is in stem cell biology and I do not have the expertise to review the computational / mathematical elements so have focussed on their biological application and how they are communicated to a biological audience. From this perspective, while it is clear that the ABC analysis approach works well for parameter derivation, some additional explanation and analysis of different types of data may be needed to make the case that this approach represents a significant improvement on existing approaches with broad applicability.

Main Comments

From the biological perspective the argument for the use of the ABC analysis approach could be significantly strengthened by demonstrating its application to a more diverse range of datasets.

For much of the data analysed, all three approaches seem to be capable of generating parameter sets within similar ranges. As the authors state for Figure 2: “All approaches broadly agreed on the estimated parameters for both experimental and synthetic data”. While the new parameterisation approaches may be more accurate from a mathematical point of view, the significance of the advance from a biological perspective could be more clearly spelled out. Would other datasets or other parameter combinations from different systems provide a more stringent test resulting in more substantial differences in parameter output between the three approaches? It would be helpful if the authors could include examples either of experimental data from the literature or simulations where this approach would substantially change a conclusion or allow a subtle but important difference to be identified.

The Discussion notes the importance of lineage analysis as a tool to explore not only normal homeostasis but also ageing, preneoplasia and cancer. The biological importance of this paper depends on the broad applicability of the approach to such situations. Given that in each of these cases fundamental assumptions used to constrain the committed progenitor model break down (for example the constraint of balanced symmetric divisions) would the ABC approach still work as well as it appears to for homeostasis? For this paper to be of broad use in the field it would be helpful to include analysis of a non-homeostatic dataset and demonstrate that ABC remains the best approach. Possible examples would include in wound healing (Aragona et al., 2017) or when clones have a growth advantage (Alcolea et al., 2014).

Similarly, the authors comment that the approach could be applied to other tissues, would this only be tissues known to observe committed progenitor dynamics or could the approach be applied to tissues with different stem/progenitor dynamics or constraints such as the intestinal epithelium?

Minor Points

Stem cell and progenitor cell are used interchangeably throughout e.g. from the Abstract “(the ‘single progenitor model’) has been argued to reproduce the observed stem cell dynamics accurately”. The use of these terms should be clarified early in the introduction to avoid confusion with alternative stem/TA, stem/committed progenitor models.

The referencing of the introduction could be more comprehensive, particularly in acknowledging the range of studies from other groups that have employed clonal lineage tracing in a range of systems.

In Figure 3C it is not clear from the plot that the final population of parameter sets overlaps the true values, would a zoomed in view on a narrower parameter space in the main figure make this clearer?

Figure 4 and related text show that individual time points result in broad parameter sets that are prone to chance variation based on 2-3 biological replicates. In terms of experimental design, would increasing the number of replicates at a single time point alone be sufficient to overcome this problem. For example, sampling 15 mice at 42 days rather than 5 mice each at 3 timepoints. In other words – is it the total number of timepoints that is critical here (as the text suggests) or the total biological replicates?

Given that the ABC approach is being proposed as the best way to analyse clonal datasets in the future, even though (as the authors state) distribution shape is the focus, why is ABC not used in Figure 4?

In Figure 4 it may be helpful to have the simulated parameter values marked on the plots as in previous figures to help orient the reader when comparing between panels.

In Figure 4e, showing that a combination of early and late timepoints is optimal for constraining parameters, 3 days is used as the early timepoint. How is the utility of this timepoint reconciled with earlier figures showing the MLE approach to be particularly inaccurate at early timepoints?

How was the biological variability parameter used in simulations calculated? Was this derived from experimental data?

In the discussion it would be useful to comment on the discrepancy between parameters derived from lineage tracing and live imaging, which has suggested an uncommitted progenitor model with any given basal cell have a 50% chance of dividing or differentiating (Rompoulas et al., Science 2016), consistent with earlier work based on nucleoside analogue labelling (Marques-Pereira and Leblond, Am J Anat 1965).

Decision letter (RSOS-202231.R0)

Dear Dr Hall

On behalf of the Editors, we are pleased to inform you that your Manuscript RSOS-202231 "Methods for analysing lineage tracing datasets" has been accepted for publication in Royal Society Open Science subject to minor revision in accordance with the referees' reports. Please find the referees' comments along with any feedback from the Editors below my signature.

Please submit your revised manuscript and required files (see below) no later than 7 days from today's (ie 02-Mar-2021) date. Note: the ScholarOne system will 'lock' if submission of the revision is attempted 7 or more days after the deadline. If you do not think you will be able to meet this deadline please contact the editorial office immediately.

on behalf of Dr Andrew Angel (Associate Editor) and Catrin Pritchard (Subject Editor)
openscience@royalsociety.org

Associate Editor Comments to Author (Dr Andrew Angel):

Comments to the Author:

The reviewers have expressed a positive opinion of the manuscript, both recommending minor revisions.

Of particular note is one reviewer's request that the discussion of the relevance of the studied approach to biologists be enhanced.

I am recommending that the manuscript undergo minor revisions in line with the reviewers' comments before it can be reconsidered for publication.

Reviewer comments to Author:

Reviewer: 1

Comments to the Author(s)

This work compares three different methods of determining the parameters of the "single progenitor model" from clone size distributions. The authors use both published and synthetic data to argue that a Bayesian-based method is optimal in terms of time and accuracy.

The problem is very interesting and timely, and the results are convincing and relevant. The manuscript is very well written, all algorithms, assumptions, and conclusions are clearly stated. I therefore recommend the manuscript for publication following a minor review.

Questions:

- Maximum likelihood method is in principle Bayesian inference with uniform prior. Could the authors briefly comment on the fundamental difference between the MLA and SMC-ABC approaches described in the manuscript?

- Jupyter notebook: I managed to get the code running, but the run time is very long and consequently it is not easy to explore the meaning of various parameters in the code. Would it be possible to add a simple demonstration that would run no longer than a minute?

Reviewer: 2

Comments to the Author(s)

In this paper the authors explore alternative strategies for the analysis of clonal lineage tracing data focussing on the previously described committed progenitor cell model. They use a combination of experimental and simulated data to test the accuracy and efficiency of three approaches allowing them to make recommendations for future analyses. My expertise is in stem cell biology and I do not have the expertise to review the computational / mathematical elements

so have focussed on their biological application and how they are communicated to a biological audience. From this perspective, while it is clear that the ABC analysis approach works well for parameter derivation, some additional explanation and analysis of different types of data may be needed to make the case that this approach represents a significant improvement on existing approaches with broad applicability.

Main Comments

From the biological perspective the argument for the use of the ABC analysis approach could be significantly strengthened by demonstrating its application to a more diverse range of datasets.

For much of the data analysed, all three approaches seem to be capable of generating parameter sets within similar ranges. As the authors state for Figure 2: “All approaches broadly agreed on the estimated parameters for both experimental and synthetic data”. While the new parameterisation approaches may be more accurate from a mathematical point of view, the significance of the advance from a biological perspective could be more clearly spelled out.

Would other datasets or other parameter combinations from different systems provide a more stringent test resulting in more substantial differences in parameter output between the three approaches? It would be helpful if the authors could include examples either of experimental data from the literature or simulations where this approach would substantially change a conclusion or allow a subtle but important difference to be identified.

The Discussion notes the importance of lineage analysis as a tool to explore not only normal homeostasis but also ageing, preneoplasia and cancer. The biological importance of this paper depends on the broad applicability of the approach to such situations. Given that in each of these cases fundamental assumptions used to constrain the committed progenitor model break down (for example the constraint of balanced symmetric divisions) would the ABC approach still work as well as it appears to for homeostasis? For this paper to be of broad use in the field it would be helpful to include analysis of a non-homeostatic dataset and demonstrate that ABC remains the best approach. Possible examples would include in wound healing (Aragona et al., 2017) or when clones have a growth advantage (Alcolea et al., 2014).

Similarly, the authors comment that the approach could be applied to other tissues, would this only be tissues known to observe committed progenitor dynamics or could the approach be applied to tissues with different stem/progenitor dynamics or constraints such as the intestinal epithelium?

Minor Points

Stem cell and progenitor cell are used interchangeably throughout e.g. from the Abstract “(the ‘single progenitor model’) has been argued to reproduce the observed stem cell dynamics accurately”. The use of these terms should be clarified early in the introduction to avoid confusion with alternative stem/TA, stem/committed progenitor models.

The referencing of the introduction could be more comprehensive, particularly in acknowledging the range of studies from other groups that have employed clonal lineage tracing in a range of systems.

In Figure 3C it is not clear from the plot that the final population of parameter sets overlaps the true values, would a zoomed in view on a narrower parameter space in the main figure make this clearer?

Figure 4 and related text show that individual time points result in broad parameter sets that are prone to chance variation based on 2-3 biological replicates. In terms of experimental design, would increasing the number of replicates at a single time point alone be sufficient to overcome this problem. For example, sampling 15 mice at 42 days rather than 5 mice each at 3 timepoints. In other words - is it the total number of timepoints that is critical here (as the text suggests) or the total biological replicates?

Given that the ABC approach is being proposed as the best way to analyse clonal datasets in the future, even though (as the authors state) distribution shape is the focus, why is ABC not used in Figure 4?

In Figure 4 it may be helpful to have the simulated parameter values marked on the plots as in previous figures to help orient the reader when comparing between panels.

In Figure 4e, showing that a combination of early and late timepoints is optimal for constraining parameters, 3 days is used as the early timepoint. How is the utility of this timepoint reconciled with earlier figures showing the MLE approach to be particularly inaccurate at early timepoints?

How was the biological variability parameter used in simulations calculated? Was this derived from experimental data?

In the discussion it would be useful to comment on the discrepancy between parameters derived from lineage tracing and live imaging, which has suggested an uncommitted progenitor model with any given basal cell have a 50% chance of dividing or differentiating (Rompoulas et al., Science 2016), consistent with earlier work based on nucleoside analogue labelling (Marques-Pereira and Leblond, Am J Anat 1965).

===PREPARING YOUR MANUSCRIPT===

If you have been asked to revise the written English in your submission as a condition of publication, you must do so, and you are expected to provide evidence that you have received language editing support. The journal would prefer that you use a professional language editing service and provide a certificate of editing, but a signed letter from a colleague who is a native

speaker of English is acceptable. Note the journal has arranged a number of discounts for authors using professional language editing services (<https://royalsociety.org/journals/authors/benefits/language-editing/>).

===PREPARING YOUR REVISION IN SCHOLARONE===

-- If you have uploaded ESM files, please ensure you follow the guidance at <https://royalsociety.org/journals/authors/author-guidelines/#supplementary-material> to include a suitable title and informative caption. An example of appropriate titling and captioning

may be found at https://figshare.com/articles/Table_S2_from_Is_there_a_trade-off_between_peak_performance_and_performance_breadth_across_temperatures_for_aerobic_sc_ope_in_teleost_fishes_/3843624.

Author's Response to Decision Letter for (RSOS-202231.R0)

See Appendix A.

Decision letter (RSOS-202231.R1)

Dear Dr Hall,

It is a pleasure to accept your manuscript entitled "Methods for analysing lineage tracing datasets" in its current form for publication in Royal Society Open Science. The comments from the Editors are included at the foot of this letter.

on behalf of Dr Andrew Angel (Associate Editor) and Catrin Pritchard (Subject Editor)
openscience@royalsociety.org

Associate Editor Comments to Author (Dr Andrew Angel):

I believe that the authors have adequately addressed the majority of the revisions suggested by the reviewers. Therefore, I am now recommending that the manuscript be accepted as is.

Appendix A

Reviewer: 1

Maximum likelihood method is in principle Bayesian inference with uniform prior. Could the authors briefly comment on the fundamental difference between the MLA and SMC-ABC approaches described in the manuscript?

The reviewer raises an interesting question. The advantages of SMC-ABC are principally flexibility and efficiency. Maximum likelihood approaches either require an analytical solution, which may not be available, or intensive simulation to produce an accurate estimation of the probabilities of different observations. These probabilities are combined with observations to determine the likelihood of different parameter combinations.

In contrast, the SMC-ABC approach applied here starts by picking sets of parameters from a uniform prior, before simulating the observed data, and measuring the similarity of the simulation outputs to the observations. New sets of parameters are generated iteratively from the old sets, rejecting parameters below a threshold that is progressively reduced as new sets are generated. This obviates the need for analytical solutions or large-scale simulations. Moreover, the generation of new parameter sets from previously collected sets effectively enriches the search in high posterior probability regions. This overcomes the inefficiencies of sampling parameter values solely from the prior distribution, which do not take account of previously accepted sampled values and the data.

We added notes about this difference in our revised introduction that explicitly point out the fundamental difference and advantage of SMC-ABC approach (page 4, lines 96-98).

Jupyter notebook: I managed to get the code running, but the run time is very long and consequently it is not easy to explore the meaning of various parameters in the code. Would it be possible to add a simple demonstration that would run no longer than a minute?

To address this concern of the reviewer we have added a new “demo” notebook that performs a substantially simplified search. Whilst this should not be used for production analysis, it illustrates how to run the analysis and the outputs and takes roughly 5 minutes to run.

Reviewer: 2

From the biological perspective the argument for the use of the ABC analysis approach could be significantly strengthened by demonstrating its application to a more diverse range of datasets.

For much of the data analysed, all three approaches seem to be capable of generating parameter sets within similar ranges. As the authors state for Figure 2: “All approaches broadly agreed on the estimated parameters for both experimental and synthetic data”. While the new parameterisation approaches may be more accurate from a mathematical point of view, the significance of the advance from a biological perspective could be more clearly spelled out. Would other datasets or

other parameter combinations from different systems provide a more stringent test resulting in more substantial differences in parameter output between the three approaches? It would be helpful if the authors could include examples either of experimental data from the literature or simulations where this approach would substantially change a conclusion or allow a subtle but important difference to be identified.

We thank the reviewer for raising this important point. There are two major contributions from this work. Firstly, previous analyses required the use of supercomputing time to estimate parameters and uncertainty, and frequently necessitated reimplementing code. Our presented approach can be performed with a desktop computer and can be adapted with little effort from the notebooks.

Secondly, the accurate estimation of uncertainties in the data is important in a wide range of fields, but specifically in the field of stem cell homeostasis there are active debates around the fundamental processes that underpin maintenance of squamous epithelial tissues. Furthermore, whilst there are multiple experimental techniques that exist to explore stem cell dynamics in the tissue, their analysis has led to different parameter estimates. Indeed, the reviewer raises the differences between parameters calculated from live imaging and lineage tracing. In this context, an accurate treatment of uncertainty could aid comparison between the outcomes of methods and resolve whether a given experiment is able to reject a hypothesis, or indeed whether two experiments give inconsistent results.

This has further impacts on the comparison of calculated parameters with historical results. The accurate estimation of uncertainties is broader, but still allow us to reject a random cell fate as proposed by Marques-Pereira and Leblond, 1965. As a further illustration of this point, we highlight in the text the result of Mascre et al., 2012 which rejects a single progenitor model based on a single late timepoint generated from 2 animals. Our presented results highlight the dangers of drawing strong conclusions from such a small sample.

To make this clearer, we have added notes to the discussion raising and expanding on these points (pages 10-11, lines 233-243).

The Discussion notes the importance of lineage analysis as a tool to explore not only normal homeostasis but also ageing, preneoplasia and cancer. The biological importance of this paper depends on the broad applicability of the approach to such situations. Given that in each of these cases fundamental assumptions used to constrain the committed progenitor model break down (for example the constraint of balanced symmetric divisions) would the ABC approach still work as well as it appears to for homeostasis? For this paper to be of broad use in the field it would be helpful to include analysis of a non-homeostatic dataset and demonstrate that ABC remains the best approach. Possible examples would include in wound healing (Aragona et al., 2017) or when clones have a growth advantage (Alcolea et al., 2014).

Similarly, the authors comment that the approach could be applied to other tissues, would this only be tissues known to observe committed progenitor dynamics or could

the approach be applied to tissues with different stem/progenitor dynamics or constraints such as the intestinal epithelium?

The reviewer raises a number of important points here. We acknowledge the importance of applying the method to diverse parameter sets to test the method, in order to establish its broad applicability. Indeed, our method can be applied to a wide range of datasets where cohesive clones are observed.

With regards to the suggestion of achieving more substantial differences between the three methods, we highlight as discussed above that MLE methods underestimate confidence intervals (CI). Underestimation of CI is a fundamental issue that can undermine our ability to analyse the data, inappropriately rejecting different compatible parameters.

We agree with the reviewer that non-neutral datasets are an interesting and important target for analysis. This is however beyond the scope of this manuscript due to specific technical issues. Non-homeostatic systems rely on an additional parameter (δ), and there is a well-established relationship between δ and r that limits our ability to distinguish them from one another.

Lastly, on the point whether our method can be applied to tissues proposed to be maintained by alternative stem cell models, we agree that SMC-ABC can indeed be applied to such systems. This would involve modifying the code to follow the underlying model assumptions for that given dataset.

We have added notes covering these points to the discussion (page 11, lines 247-249).

Stem cell and progenitor cell are used interchangeably throughout e.g. from the Abstract “(the ‘single progenitor model’) has been argued to reproduce the observed stem cell dynamics accurately”. The use of these terms should be clarified early in the introduction to avoid confusion with alternative stem/TA, stem/committed progenitor models.

We would like to thank the reviewers for raising this potentially confusing language. Given that these terms are used in alternative models in the literature, we have replaced the term “stem cell” with “progenitor cell” throughout our revised manuscript.

The referencing of the introduction could be more comprehensive, particularly in acknowledging the range of studies from other groups that have employed clonal lineage tracing in a range of systems.

We have extended the references used in the introduction to include a broader range of studies performed lineage tracing (Mascré et al., 2012, Rompolas et al., 2016, Sada et al., 2016) (page 2, line 39).

In Figure 3C it is not clear from the plot that the final population of parameter sets overlaps the true values, would a zoomed in view on a narrower parameter space in the main figure make this clearer?

We would like to thank the reviewer for pointing this out. In response to this, one additional image has been added in Figure 3C with a narrower parameter space including the parameter estimation areas and the true values.

Figure 4 and related text show that individual time points result in broad parameter sets that are prone to chance variation based on 2-3 biological replicates. In terms of experimental design, would increasing the number of replicates at a single time point alone be sufficient to overcome this problem. For example, sampling 15 mice at 42 days rather than 5 mice each at 3 timepoints. In other words – is it the total number of timepoints that is critical here (as the text suggests) or the total biological replicates?

The reviewer raises an interesting point. Analysis of both experimental and synthetic datasets has shown that likelihood distributions at late timepoints become highly similar. At the same time, late timepoints also illustrate some of the key properties of the system, such as the linear growth of average clone sizes over time, which are key pieces of evidence in favour of the single progenitor model. Early timepoints in contrast have distinct, but broad likelihood distributions. This supports parameter inference when analysed together, but our results show that a single timepoint would be insufficient. As such the design of the experiments should take account of both the overarching features we expect to find from the model and the need to parameterise accurately to be confident in both the fundamental processes driving stem cell maintenance of tissues and their underlying parameters. This leads us to conclude that increasing sampling over fewer timepoints, spread across the range of times, is the best approach.

In response to the reviewers specific comment, we believe that a single extremely well sampled timepoint would not be enough for accurate parameterisation and a combination of early and late timepoints with increased biological sampling is recommended for future experimental design. Moreover, we would stress that in our view that one risk to the researcher studying stem cell growth arises from the overinterpretation of underpowered timepoints rather than simply in the ability to resolve parameters. We have expanded our notes on experimental design to take account of this (page 9, lines 202-205).

Given that the ABC approach is being proposed as the best way to analyse clonal datasets in the future, even though (as the authors state) distribution shape is the focus, why is ABC not used in Figure 4?

The purpose of this analysis was to investigate the contribution of individual timepoints to the likelihood distribution, to explore whether it is safe to rely on likelihood results drawn from individual timepoints, and how the likelihood distribution varies over time. This is important considering the fact that previous studies made deductions based on individual timepoints (Mascreé et al. 2012). As we are interested in an accurate estimation of the shape of the distribution, we applied MLE with an analytical solution to synthetic data with an exponential distribution. Whilst this may underestimate uncertainty, the variations of shape will remain meaningful. Our text has been updated to note this (page 8, lines 177-180).

In Figure 4 it may be helpful to have the simulated parameter values marked on the plots as in previous figures to help orient the reader when comparing between panels.

We would like to thank the reviewer for pointing this out. To address this, we added the simulated parameter values on the plot.

In Figure 4e, showing that a combination of early and late timepoints is optimal for constraining parameters, 3 days is used as the early timepoint. How is the utility of this timepoint reconciled with earlier figures showing the MLE approach to be particularly inaccurate at early timepoints?

We thank the reviewer for raising this point. Early timepoints can be inaccurate if the cell cycle distribution is not taken account of, and indeed as noted above no one timepoint should be used in isolation for parameter estimation. The purpose of this analysis was however to show how the shape of the likelihood distribution can differ when analysing single timepoints in contrast to analysing all timepoints together, and synthetic data was generated using an exponential distribution so the early timepoints would be accurate for the method used. As such, that early timepoints can be inaccurate when cell cycle distribution times are not known does not undermine our argument that for accurate parameterisation we need a range of well sampled timepoints.

How was the biological variability parameter used in simulations calculated? Was this derived from experimental data?

For generating synthetic datasets that incorporate biological variability we chose parameters in broad agreement with previously inferred values. We have added notes on this in the results section (page 7, lines 142-144).

In the discussion it would be useful to comment on the discrepancy between parameters derived from lineage tracing and live imaging, which has suggested an uncommitted progenitor model with any given basal cell have a 50% chance of dividing or differentiating (Rompoulas et al., Science 2016), consistent with earlier work based on nucleoside analogue labelling (Marques-Pereira and Leblond, Am J Anat 1965).

We have added notes to this effect in the discussion (page 11, lines 240-243).